# Adenovirus E1B-55K regulates p53-dependent and -independent gene expression during infection

Laura Seddar, Konstantin von Stromberg, Luca D. Bertzbach, Adam Grundhoff, Thomas Dobner, Wing Hang Ip*

Leibniz Institute of Virology, Hamburg, Germany

* winghang.ip@leibniz-liv.de

## Abstract

Human adenoviruses (HAdVs) are widespread pathogens with the capacity to manipulate host cellular pathways, including critical tumor suppressor networks. During oncogenic cell transformation, the adenoviral E1B-55K protein serves as a multifunctional viral regulator that, inter alia, modulates both p53-dependent and -independent pathways – though this function has been disputed in the context of viral infection. Here, we elucidate the dual role of E1B-55K in disrupting host defenses, focusing on its impact on p53 signaling and interferon-stimulated genes (ISGs) during infection. Using RNA-seq and follow-up experimental validation in A549 (p53 wildtype) and H1299 (p53-null) cells infected with wildtype HAdV-C5 or an E1B-55K-deficient mutant, we show that E1B-55K suppresses p53-mediated transcriptional responses. Concurrently, E1B-55K modulates ISG expression in a context-dependent manner. Our results reveal that E1B-55K leverages cellular context to optimize viral replication by targeting a host tumor suppressor and indicate interference with innate immune pathways. Our study thereby uncovers a previously underappreciated aspect of E1B-55K function during infection, offering insights into its repressive activity and solidifying its role as a multifunctional viral oncoprotein with broader implications for the HAdV replication cycle.

## Author summary

In this study, we investigated how the multifunctional adenoviral protein E1B-55K manipulates host defense mechanisms to promote productive infection. E1B-55K is best known for blocking the tumor suppressor p53. Here, we discovered that its influence extends much further. We found that this protein also interferes with type I interferon responses, which normally act as powerful barriers against viral infections. By dampening both p53 activity and antiviral signaling, E1B-55K creates a cellular environment that favors viral replication. In the absence of

**Data availability statement:** All raw sequencing datasets used in this study are available at the European Nucleotide Archive (https://www.ebi.ac.uk/ena) with accession number PRJEB89469.

**Funding:** This research was conducted as part of a project supported by funding from the Wilhelm Sander-Stiftung (grant number R2024.123.1 to TD and WHI). The LIV is supported by the Freie und Hansestadt Hamburg and the German Bundesministerium für Gesundheit. The funders had no role in study design, data collection and analysis, decision to publish, or preparation of the manuscript.

**Competing interests:** The authors have declared that no competing interests exist.

E1B-55K, cells mounted stronger p53 and interferon-driven responses, which together restricted viral replication. Our findings show that a single viral protein can help reprogram multiple host defense pathways, drawing interesting parallels with how other cancer-driving genes disrupt cellular control measures. By uncovering these diverse functions of E1B-55K, we provide new insights into adenovirus infection and reveal novel links between this viral oncogene and the host defense system.

## Introduction

Human adenoviruses (HAdVs) rely on the expression of early viral genes to reprogram host cell pathways and establish a cellular environment conducive to viral replication. These early gene products are critical for evading both innate and adaptive immune defenses [1]. In particular, they antagonize two major antiviral barriers: the type I interferon (IFN) response – which induces a broad array of interferon-stimulated genes (ISGs) – and the DNA damage response (DDR), which includes activation of the tumor suppressor p53, leading to cell cycle arrest and apoptosis [2–4]. The HAdV early proteins, notably E1A and E1B-55K, are essential in counteracting these responses. E1A inhibits ISG transactivation through multiple distinct mechanisms that disrupt key components of the antiviral signaling cascade and transcriptional regulation [5,6]. Additionally, it interferes with the stability and function of antiviral transcription factors, ultimately dampening the host innate immune response [7]. By sequestering pRb and releasing E2F to ensure S-phase entry, E1A inadvertently stabilizes the tumor suppressor, inducing apoptosis [8–10]. Thus, p53 activation presents a secondary barrier that HAdVs must overcome. In this context, E1B-55K inhibits p53 by forming a complex with the viral E4orf6 protein, promoting p53 ubiquitination and proteasomal degradation. Additionally, E1B-55K can sequester p53 into the cytoplasm, preventing its transcriptional activity in the nucleus [11,12].

Beyond its role during productive infection, E1B-55K also plays a central role in oncogenic cell transformation [11–13]. It has been widely established that E1B-55K acts as a transcriptional repressor of p53-regulated genes to efficiently counter growth arrest induced by E1A, although the exact mechanisms remain poorly understood [14–18]. Our recent work demonstrated that E1B-55K not only represses p53 targets but also members of the AP-1 and TEAD/YAP/TAZ transcription factor complexes during transformation [19]. Conversely, studies in the context of viral infection suggest that E1B-55K primarily controls ISG expression rather than p53 targets [20,21]. The repressive activities of E1B-55K therefore may be context-specific in transformed versus infected cells. While E1B-55K is well known for repressing p53-regulated genes, its broader effects on host transcription remain unclear. Interestingly, increasing evidence suggests that p53 is also induced during viral infections as a downstream transcriptional target of IFN signaling [22]. Therefore, we sought to investigate whether (i) E1B-55K also regulates host gene

expression during infection, (ii) this regulation involves crosstalk between p53 and IFN, and (iii) these changes may influence viral gene expression and productive replication.

In this study, we combined transcriptome analyses with experimental validation in p53-positive A549 and p53-negative H1299 cells infected with HAdV-C5 and E1B-55K-deficient mutant (ΔE1B-55K) to investigate E1B-55K-dependent gene expression patterns during infection and in the context of IFN-alpha (IFN-α) stimulation. Our findings show that E1B-55K effectively antagonizes p53 signaling in A549 cells. Cells with an activated type I IFN response display reduced viral replication in the absence of E1B-55K. Additionally, E1B-55K moderately contributes to the suppression of ISG expression in A549 but not H1299 cells. These results underscore the dual role of E1B-55K in modulating innate immune and tumor suppressor pathways to facilitate HAdV replication.

## Materials and methods

### Cells and culture conditions

A549 (DMSZ ACC 107; German Collection of Microorganisms and Cell Cultures), H1299 (ATCC CRL-5803; American Type Culture Collection) and primary human foreskin fibroblasts (HFF) were maintained as monolayers in Dulbecco's Modified Eagle Medium (DMEM; Gibco) supplemented with 10% fetal bovine serum (Sigma Aldrich) and 1% penicillin/streptomycin solution (10,000 U/ml penicillin; 10 mg/ml streptomycin in 0.9% NaCl, PAN Biotech) at 37°C and 5% $CO_2$. Cells were regularly tested for mycoplasma contaminations using the PCR Mycoplasma Test Kit I/C (PromoKine). HFFs were maintained in monolayer cultures for no more than 13 passages.

### Viruses and infections

HAdV-C5 wildtype virus with HA-tagged E1B-55K was derived from the wildtype reference strain H5pg4100 [23], which lacks a 1863 bp stretch of the E3 region (nt 28602-30465), and was generated via homologous recombination by λRed proteins as described earlier (S1 Table) [24,25]. The insertion was verified by analyzing restriction digestion patterns and Sanger sequencing. The ΔE1B-55K virus has been published previously [26]. Both viruses were propagated in A549 cells and titrated in A549 and H1299 cells exactly as previously described [23]. Universal type I IFN (human IFN-α hybrid protein; PBL Assay Science) was diluted to 1000 U/µl in sterile PBS containing 0.1% (w/v) BSA and aliquots were stored at -80°C. Cells were pretreated with 500 U/ml of IFN-α or vehicle only (PBS + 0.1% BSA) for 16–18 h prior to infection, after determination of the optimal concentration via cell viability assays (S1 Fig). Infection was followed by incubation in medium containing IFN-α or vehicle only (MOI 30 for tumor cell lines A549 and H1299; MOI 100 for HFFs). For virus yield experiments, pretreated and infected cells were harvested 24 hpi and lysed by three freeze and thaw cycles. Serial dilutions of virus containing supernatants were used for re-infection of the same cell line and virus titers were assessed by immunofluorescent staining for the adenovirus DNA-binding protein (DBP).

### Western blotting

Cells were lysed in radioimmunoprecipitation assay (RIPA) buffer (50 mM Tris-HCl pH 8.0, 150 mM NaCl (Fluka), 5 mM EDTA (Merck), 1 mM dithiothreitol (DTT), 0.1% sodium dodecyl sulfate, 1% Nonidet P-40, 0.1% Triton X-100 (Sigma-Aldrich), 0.5% sodium deoxycholate) freshly supplemented with 1% phenylmethylsulfonyl fluoride (PMSF, Sigma-Aldrich), 0.1% aprotinin, 1 µg/ml leupeptin (Roche), 1 µg/ml pepstatin (Biomol) at 4°C. Lysates were sonicated and insoluble debris was pelleted. Protein concentrations were measured with a Bradford Reagent-based protein assay (BioRad) and samples were mixed with 5x SDS sample buffer. Equal amounts of total protein were separated by SDS-polyacrylamide gel electrophoresis and transferred onto 0.45 µm nitrocellulose membranes (Amersham). Membranes were blocked overnight at 4°C in phosphate-buffered saline (PBS) containing 5% non-fat dry milk powder. Next, membranes were washed in PBS supplemented with 0.1% Tween 20 (PBS-Tween), before they were

incubated with the respective primary antibody dilutions at 4°C for 2 h (S2 Table). Afterwards, membranes were washed as described and incubated with the corresponding HRP-conjugated secondary antibodies (Jackson ImmunoResearch) diluted 1:10,000 in PBS-Tween containing 3% non-fat dry milk powder for 2 h at 4°C. After three final wash steps, proteins were visualized using the SuperSignal West Pico Chemiluminescent Substrate (Thermo Scientific) on medical X-ray films (CEA RP) or with the ChemoStar Plus (Intas) imaging system. Autoradiograms were scanned and cropped using Adobe Photoshop CS6 and figures were prepared in Inkscape 1.2.

### RNA extraction

$0.5\text{-}1 \times 10^6$ cells were resuspended in 1 ml TRIzol reagent (Thermo Fisher) and RNA was isolated according to the manufacturer's instructions. Here, 3 µl of 5 mg/ml glycogen (Invitrogen) was used to increase recovery. Additionally, samples were washed twice with ethanol to reduce phenol carryover. Lastly, samples were eluted in an appropriate volume of DEPC-$H_2$O (Qiagen) and stored at -80°C.

### cDNA synthesis

Equal quantitates of RNA were reverse transcribed using the High Capacity cDNA Reverse Transcription Kit (Applied Biosystems) according to the manufacturer's instructions. Obtained cDNA was diluted 1:100 in DEPC-$H_2$O and quantified with the SensiMix SYBR Hi-ROX Kit (Meridian Bioscience).

### RNA-seq

Prior to library preparation, RNA integrity was assessed utilizing the Agilent 2100 Bioanalyzer System combined with a RNA 6000 Nano Chip (Agilent). Polyadenylated (poly(A)) mRNA fractions were purified from the total RNA and RNA-seq libraries were generated using the CORALL mRNA-Seq V2 Library Prep Kit (Lexogen) according to the manufacturer's recommendations. The concentrations and sizes of the final cDNA libraries were measured with the RNA High Sensitivity Chip on an Agilent 2100 Bioanalyzer System (Agilent). All samples were normalized to 2 nM and pooled at equimolar concentrations. The library pool was sequenced on a NextSeq 2000 (Illumina) using paired-end sequencing. All samples were sequenced at the high-throughput sequencing technology platform of the LIV.

### DNA extraction

Cell pellets from mock-, wildtype- and ΔE1B-55K-infected A549 and H1299 cells were harvested at 24 hpi. Total DNA was isolated using the QIAamp DNA mini kit (Qiagen) according to the manufacturer's instructions. Viral DNA was quantified from equal concentrations (45 ng/µl) of total DNA via qPCR with primers targeting the viral gene L4-100K (S1 Table).

### qPCR

cDNA or viral DNA were quantified using the SensiMix SYBR Hi-ROX Kit (Meridian Bioscience). Real time thermocycling was performed on the Rotor Gene 6000 (Corbett Research). Ct values were corrected for different amplification efficiencies using standard curves for each primer pair. Samples were analyzed with the ΔΔCt method using GAPDH as the internal reference. All runs were performed in technical replicates for three independent experiments. Primer pairs used for cDNA or viral DNA quantification are listed in S1 Table.

### Cell viability assay

The CellTiter-Glo Luminescent Cell Viability Assay (Promega) was used according to the manufacturer's instructions to determine the number of viable cells upon pretreatment with ranging IFN-α concentrations and after infection with the respective virus. Luminescence was measured on the Spark multimode microplate reader (Tecan).

## Data analyses

RNA-seq expression data was generated by quantifying paired-end mRNA reads using a decoy-aware GRCH38 (Ensembl 105) transcriptome with the salmon quantifier software [27] via the selective alignment algorithm. This transcriptome was generated by concatenating the respective non-coding- and coding RNA in front of the Human genome assembly, which generates a combined reference file that is subsequently used by salmon to generate an index. This workflow can be obtained from the Zenodo digital library (10.5281/zenodo.15389108). The differential gene expression of the quantification data was analyzed with the R-based DESeq2 package according to the developer's vignette [28]. The script utilized here can be obtained from the Zenodo digital library under 10.5281/zenodo.15389112. The output of this DEG analysis and all investigated conditions can be found in S1 File. The WebgestaltR [29] package was used to provide biological context to the DEG-Analysis.

## Statistical analyses

Statistical analyses were performed using GraphPad PRISM 10.4.2 and Microsoft Excel 2016. The specific tests applied are detailed in the respective figure legends. Results were considered statistically significant at $P < 0.05$.

## Results

### E1B-55K represses p53-dependent transcription during adenoviral infection

During E1A/E1B-induced transformation, E1B-55K represses host transcription by directly interacting with transcription factors, notably inhibiting key regulatory networks like the p53 pathway – the best-characterized target to date. However, the full range of E1B-55K functions during infection remains unclear and may depend on the cellular background. In this study, we first aimed to determine whether the role of E1B-55K as a transcriptional repressor is functionally equivalent to cellular transformation and to evaluate the potential extent of its regulatory impact on host signaling pathways. Western blot analysis of infected A549 cells displayed robust expression of E1B-55K in wildtype infection at 24 hpi and absence of the protein in ΔE1B-55K infection. The presence of the early protein was accompanied by a strong reduction in the steady-state levels of Mre11 and p53, both well-known interaction partners of the E1B-55K/E4orf6 complex, and consequently the major p53-mediated target gene CDKN1A during wildtype infection (Fig 1A). However, p53 was not completely degraded in wildtype infected cells and low levels were still detectable at this late time point of infection. As expected, reduced levels of Mre11, p53 and CDKN1A were not observed in ΔE1B-55K mutant virus infected cells (Fig 1A). Furthermore, we found that the presence of E1B-55K influences the global transcriptional profile of infected cells, as reflected by an E1B-55K-dependent signature that was distinct from general infection-related changes, thus underscoring a unique regulatory role for E1B-55K (Fig 1B). The most substantial individual genes affected by E1B-55K expression were p53 target genes, as their transcription is highly repressed in the presence of the wildtype virus while transcription remains active in the ΔE1B-55K mutant (Fig 1C, marked in green; Fig 1D). These findings were reinforced at the pathway level, which revealed significant enrichment for p53-associated biological processes among downregulated genes, including apoptosis, cancer signaling, and DDR pathways, among others (Fig 1E). Our experiments indicated that E1B-55K is the primary viral protein responsible for countering the host cell p53 response in infection. Together, these data suggest that the interaction between p53 and E1B-55K plays a crucial role in modulating the p53 transcriptional response during adenoviral infection, supporting the idea that this interaction is also functionally relevant in the context of infection. Interestingly, expression of E1B-55K during virus infection was not only associated with downregulation of the p53 transcriptional network and apoptosis-related genes but also with upregulation of genes involved in oxidative phosphorylation, components of the electron transport chain in mitochondria and cytoplasmic ribosomal proteins (Fig 1E).

E1B-55K-dependent repression of host transcription was less pronounced in p53-negative H1299 cells compared to A549 cells (Fig 2A and 2B). Consistent with our previous findings from E1-transformed rat cells [19], infection with adenovirus also affected the TEAD transcriptional network, primarily in p53-negative H1299 cells. Our analysis revealed downregulation of the Hippo signaling pathway, along with reduced expression of genes activated by the TEAD transcriptional

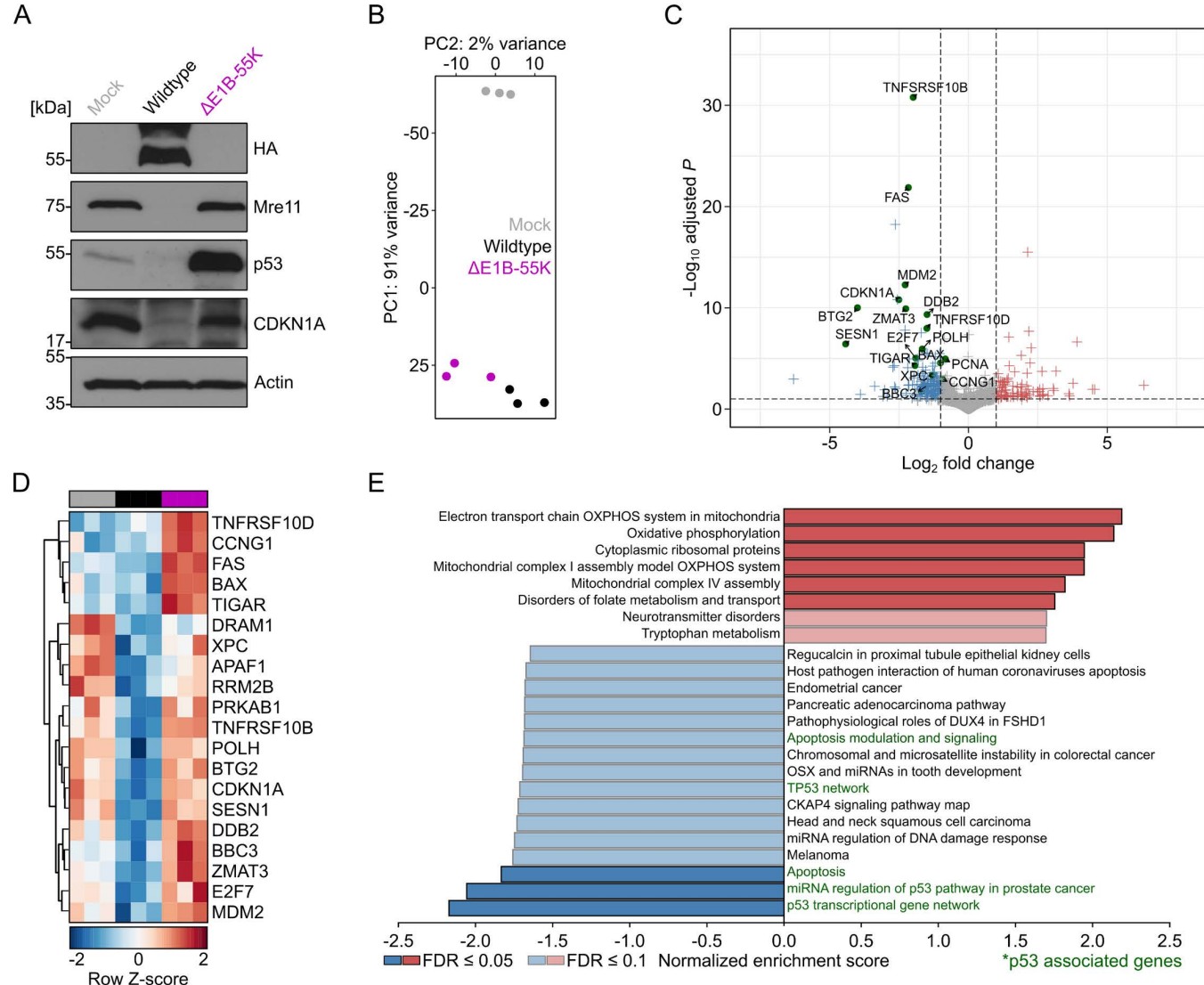

**Fig 1. E1B-55K suppresses the p53 transcriptional network during infection in A549 cells. (A)** Western blot showing the expression of E1B-55K target genes in A549 cells infected with either wildtype (HA-tagged E1B-55K-expressing) virus or a ΔE1B-55K mutant (lacking E1B-55K expression). Cells were infected with MOI 30 and analyzed after 24 hours. This representative replicate illustrates the steady-state levels of p53, Mre11 and CDKN1A during infection (antibodies are listed in S2 Table). **(B)** Principal component analysis of mock-, wildtype- and ΔE1B-55K-infected A549 cells after differential gene expression analysis. **(C)** Volcano plot comparing gene expression between wildtype- and ΔE1B-55K-infected A549 cells. Genes with an adjusted $P$-value <0.1 and a $\log_2$ fold change <-1 are shown in blue, while those with an adjusted $P$-value <0.1 and a $\log_2$ fold change >1 are shown in red. Genes belonging to the p53 transcriptional network (WikiPathways database) are highlighted with green circles. **(D)** Heatmap displaying row Z-scores of p53 pathway genes that were downregulated in (C) and are marked in green. The grey, black, and purple boxes at the top of the panel represent mock, wildtype, and ΔE1B-55K infection, respectively. **(E)** Global pathway analysis of differentially expressed genes comparing wildtype- with ΔE1B-55K-infected A549 cells using gene set enrichment analysis. The normalized enrichment scores of significantly up- and downregulated pathways are shown, with pathways meeting FDR<0.1 highlighted in light blue/red and those with FDR<0.05 in dark blue/red. p53-associated pathways are written in green. Red bars indicate pathways enriched in upregulated genes, while blue bars indicate pathways enriched in downregulated genes.

coactivators YAP and its paralog TAZ (Figs 2C and 2D and S2). Follow-up co-immunoprecipitation experiments confirmed a physical interaction between transfected E1B-55K and TEAD4 in H1299 cells, suggesting the existence of a functional E1B-55K–TEAD axis (Figs 2E and S2).

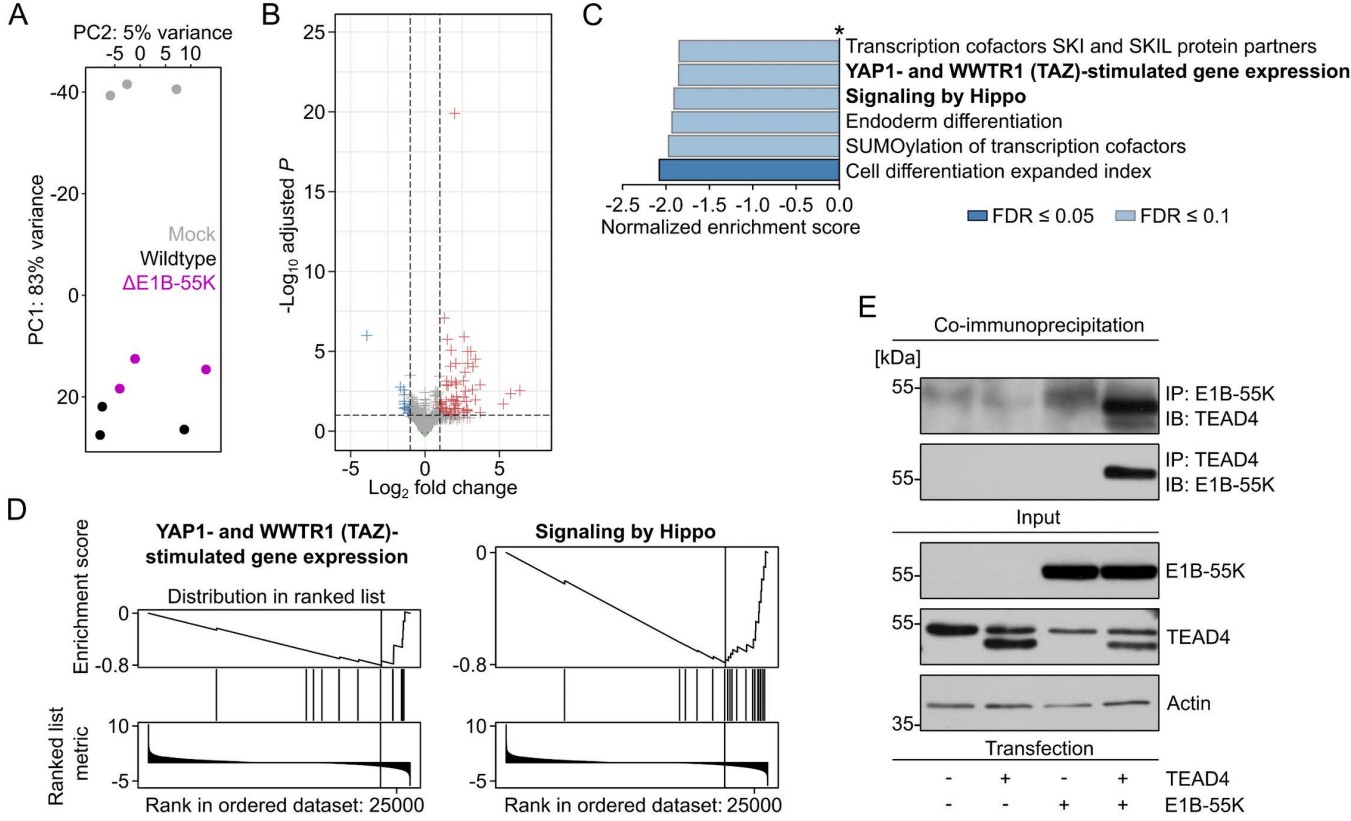

**Fig 2. Regulation of the Hippo signaling pathway by E1B-55K in p53-negative H1299 cells. (A)** Principal component analysis of mock-, wildtype- and ΔE1B-55K-infected H1299 cells after differential gene expression analysis. **(B)** Volcano plot comparing gene expression between wildtype- and ΔE1B-55K-infected H1299 cells. Genes with an adjusted $P$-value $<0.1$ and $\log_2$ fold change $<-1$ are shown in blue, while genes with an adjusted $P$-value $<0.1$ and $\log_2$ fold change $>1$ are shown in red. **(C)** Pathway analysis of up- and downregulated genes comparing wildtype- with ΔE1B-55K-infected H1299 cells using GSEA, showing the normalized enrichment score of pathways with a FDR $<0.1$ (light blue) and FDR $<0.05$ (dark blue). Shown here are only the downregulated pathways, the upregulated ones (*) can be found in S2 Fig. Blue bars indicate pathways enriched in downregulated genes. **(D)** GSEA plots of "YAP1- and WWTR1 (TAZ) stimulated gene expression" and "signaling by Hippo" pathways from (C), illustrating the distribution of included genes in the stat-metric ranked gene expression list. **(E)** Western blot displaying the expression of co-transfected E1B-55K and TEAD4 in H1299 cells. Co-immunoprecipitation in either direction is shown in the upper blots to highlight interaction between the proteins.

## p53 and IFN responses restrict replication of the ΔE1B-55K mutant

Given that E1B-55K modulates p53 levels during infection, and considering the well-established interplay between p53 signaling and the IFN response [30], we explored whether E1B-55K contributes to the manipulation of host antiviral defenses via this axis. During viral infection, pathogens commonly interfere with the IFN pathway to support efficient replication. Therefore, we pretreated A549 and H1299 cells with IFN-α before infection to assess whether IFN induction influenced adenoviral replication. Specifically, we compared viral replication kinetics in p53-positive A549 and p53-negative H1299 cells in the presence or absence of E1B-55K and IFN-α treatment. Both cell lines showed strong activation of IFN-associated pathways, indicating a functional response to treatment (S3 and S4 Figs). A schematic overview of the experimental design and workflow is provided for clarity in Fig 3A. Western blot analyses demonstrated only minor differences in the E1A, E1B-55K and DBP protein levels, thus ruling out variable early viral protein expression as the primary cause of the observed defects in virus progeny production (Figs 3B and S5). In A549 cells, the E1B-55K-deficient mutant

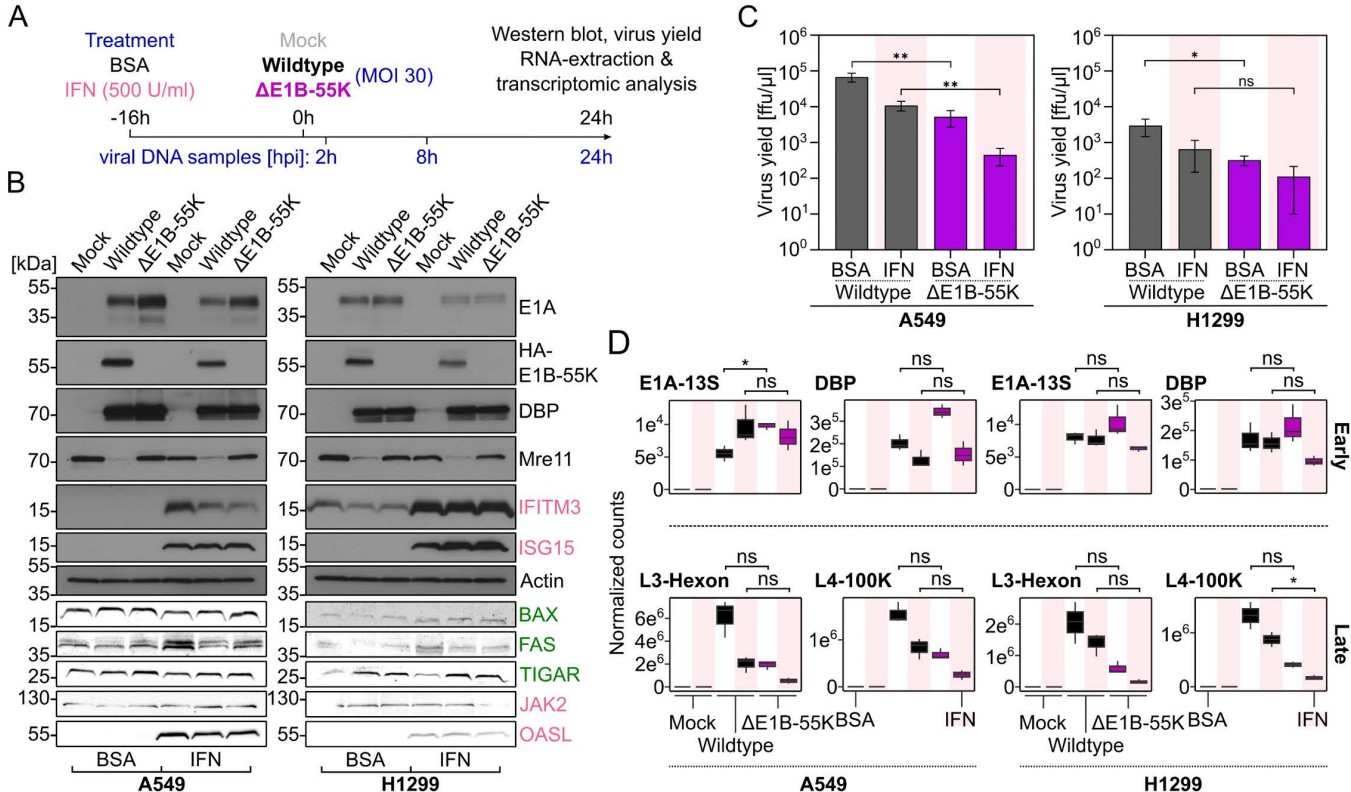

**Fig 3. Adenovirus lacking E1B-55K is more severely impacted by IFN than wildtype. (A)** Interferon-alpha induction scheme in A549 and H1299 cells. The time points post-infection at which DNA is extracted for viral DNA quantification are illustrated below the timeline bar. **(B)** Western blot from one of the sequenced replicates, displaying selected viral- and IFN-induced proteins (antibodies are listed in S2 Table). Proteins associated with the p53 pathway and the immune response are marked in green and pink, respectively. The upper seven proteins were detected on medical X-ray films and the lower five proteins were visualized via the ChemoStar Plus. Additional replicates are shown in S5 Fig. **(C)** Virus yield was determined 24 hpi with MOI 30 by anti-DBP (B6-8) immunofluorescence staining. Bar graphs represent the mean virus yield of three independent experiments, error bars indicate SD. Black and purple bars represent wildtype and ΔE1B-55K virus, respectively. Treatment with BSA or IFN is indicated with white and pink background, respectively. Statistical significance was determined using two-tailed t-test. \*\*$P$-value < 0.01, \*$P$-value < 0.05, ns $P$-value > 0.05. **(D)** Normalized viral mRNA counts of selected early (upper row) and late (lower row) genes. \*adjusted $P$-value < 0.1; ns adjusted $P$-value > 0.1. Light grey, black, and purple box plots represent mock, wildtype virus, and ΔE1B-55K virus conditions, respectively. Treatment with BSA or IFN is indicated with white and pink background, respectively.

exhibited a significantly lower virus yield than the wildtype virus, and this reduction was even more pronounced upon IFN treatment. This additive effect was also detected in H1299 cells, though it was less pronounced and not statistically significant (Fig 3C). As shown in Fig 3D, for both viruses this trend is reflected by a more pronounced IFN-induced decrease in viral late (structural) L3-Hexon transcript in A549 cells compared to H1299 cells. Notably, IFN treatment did not result in significant differences in DNA replication between the two viruses across the two distinct cell lines, although the E1B-55K deficient mutant showed slightly reduced viral DNA synthesis that was further decreased upon IFN treatment (S6 Fig). Overall, A549 cells displayed higher DNA replication efficiencies and virus yield compared to H1299 cells (Figs 3C and S6). The elevated basal levels of IFITM3 in H1299 cells combined with a more substantial IFN-induced increase in ISG15 and IFITM3 protein levels (Fig 3B), likely contribute to stronger inhibition of virus replication observed in this cell line. The observation that viral replication is generally reduced in the absence of E1B-55K and further diminished upon IFN treatment in both cell lines, suggests a dual role for E1B-55K in overcoming both p53- and IFN-mediated antiviral defenses.

## E1B-55K influences IFN signaling in addition to p53 depending on the cellular context

Next, we conducted additional transcriptomic profiling to determine whether these regulatory effects occur at the transcriptional level, particularly considering the known post-transcriptional functions of E1B-55K [12]. The previously described replication defect of the ΔE1B-55K mutant was further exacerbated by addition of exogenous IFN-α, which significantly reduced viral progeny in A549 cells (Fig 3C). Interestingly, global pathway analysis of transcriptomes from infected cells identified mild suppression of IFN-associated pathways in both cell lines when E1B-55K was present, although this effect was more pronounced in A549 cells (Fig 4A and 4B). Of note, this suppression was only detectable upon IFN stimulation and would have been missed under baseline infection conditions, as in prior studies. With the exception of ISG15, ISG expression was consistently lower in IFN-treated infected cells than in the uninfected controls. This effect was observed in both A549 and H1299 cells, regardless of E1B-55K expression (Fig 4C). It is important to mention that E1A is described as the primary repressor of the host immune response and equally present in both wildtype and ΔE1B-55K infections (Figs 3B and 3D and S5). Some individual IFN signaling genes were significantly repressed by E1B-55K in A549 cells, but not in H1299 cells (Fig 4C and 4D), albeit only on a transcriptional level, while we could not identify changes on a protein level (Figs 3B and S5). Importantly, RT-qPCR validation of selected p53 targets and ISGs downregulated in an E1B-55K-dependent manner confirmed significant repression of CDKN1A and MDM2 (Fig 4C), as well as the IFN-regulated genes JAK2 and IRF9, upon IFN treatment of infected A549 cells. Although OASL was identified as one of the most strongly downregulated IFN-inducible genes in A549 cells, this finding was not confirmed by RT-qPCR (S9 Fig). Intriguingly, in H1299 cells, IRF9 and STAT2 expression was significantly reduced under both treatments when E1B-55K was present during infection (S10 Fig). Extensive follow-up studies using a shRNA-mediated knockdown of p53 in A549 cells could not trace this effect back to that of p53 alone (S11 and S12 Figs), indicating independence of the p53 and IFN networks. In addition, in order to reproduce observations made in the aforementioned cancer cell lines, we infected IFN-treated HFF cells with wildtype or ΔE1B-55K virus, but could not identify any effect on selected ISGs on mRNA or protein level (S13 and S14 Figs). Notably, viral protein steady states were severely reduced in primary cells upon IFN-α treatment in our hands, preventing a robust comparison between viruses and treatments.

These results suggest that in A549 cells, p53 contributes to the restriction of ΔE1B-55K replication, while IFN responses represent an additional barrier that the virus fails to counteract in the absence of E1B-55K. Moreover, E1B-55K might modulate IFN responses independent of p53, which is supported by our knockdown experiments (S11 and S12 Figs), potentially via interactions with other host factors.

## Discussion

The adenoviral protein E1B-55K is primarily recognized for its role in inhibiting p53, specifically by blocking its pro-apoptotic and cell cycle arrest activities to transform cells. However, most information about E1B-55K-mediated regulation of p53 is based on studies in transformation rather than productive infection [11,12,31]. In this study, we found that HAdV-C5 E1B-55K significantly repressed canonical p53 target genes such as CDKN1A, MDM2 and FAS during productive infection of A549 cells. The majority of downregulated genes were strongly associated with p53 transcriptional networks and apoptosis signaling. Consequently, our findings reinforce the role of E1B-55K as a transcriptional repressor of p53-signaling during infection. Transcriptomic analysis of E1A/E1B-transformed BRK cells showed that E1B-55K modulates TEAD-driven transcription [19]. Notably, Hippo signaling was also disrupted in p53-deficient H1299 cells upon infection, suggesting that E1B-55K alters transcriptional programs independently of p53. This is significant, as both pathways regulate diverse cellular processes, enabling E1B-55K to impact broad functions via few targets. The Hippo pathway regulates cell growth, proliferation and homeostasis and has been shown to be critical in development, stem cell function, and tissue regeneration [32,33]. The context-dependent interplay between p53 and Hippo signaling ranges from cooperative to antagonistic [34]. Our future work will investigate this aspect, given its likely central role in E1B-55K-mediated transcriptional regulation.

**Fig 4. E1B-55K represses the IFN response in A549 but not H1299 cells. (A)** Principal component analysis of mock-, wildtype- and ΔE1B-55K-infected A549 (left) and H1299 (right) cells after differential gene expression analysis. Circles and triangles indicate BSA and IFN treatment, respectively. **(B)** Pathway analysis of downregulated genes comparing wildtype with ΔE1B-55K infection upon IFN treatment using GSEA, showing the normalized enrichment score for pathways with FDR<0.1 (light blue) and FDR<0.05 (dark blue). The upper and lower bar graphs represent pathways

enriched in infected A549 and H1299 cells, respectively. Only downregulated pathways are displayed; upregulated pathways are shown in S7 and S8 Figs. Immune response-associated pathways are highlighted in pink and p53-associated pathways in green. The dotted line separates the two cell lines, as labeled on the right. **(C)** Normalized viral mRNA counts of selected p53 and immune-system associated pathways from wildtype- and ΔE1B-55K-infected A549 (top) and H1299 (bottom) cells treated with IFN. ****adjusted $P$-value < 0.0001, ***adjusted $P$-value < 0.001, *adjusted $P$-value < 0.1, ns adjusted $P$-value > 0.1. Light grey, black, and purple box plots represent mock, wildtype virus, and ΔE1B-55K virus conditions, respectively. Treatment with BSA or IFN is indicated with white and pink background, respectively. **(D)** Volcano plot comparing gene expression between wildtype- and ΔE1B-55K-infected A549 (top) and H1299 (bottom) cells treated with IFN. Genes with an adjusted $P$-value < 0.1 and $\log_2$ fold change < 0 are colored in blue, while genes with an adjusted $P$-value < 0.1 and $\log_2$ fold change > 0 were colored in red. Genes that belong to the IFN-associated gene network are shown with pink dots.

While our study demonstrates that E1B-55K significantly represses the p53 transcriptional gene network during infection, previous microarray data generated in HFF cells revealed that E1B-55K does not affect p53-mediated gene expression. Instead, the viral protein was primarily involved in inhibiting genes associated with immune response and antiviral defense [21]. Follow-up studies further substantiated this immunomodulatory function when E1B-55K potently inhibited the adverse effects of exogenous IFN on virus replication and replication compartment formation [20]. Conversely, pretreatment of HFF cells with IFN prior to infection markedly reduced viral protein expression in this study, limiting their use for further analyses (S11 Fig). This may reflect donor variability, as the cells were freshly isolated in-house.

While p53 is well established as a tumor suppressor that prevents cell transformation, recent studies show crosstalk between p53 and IFNs, highlighting a new layer of its regulation [22]. On the one hand, activation of p53 induces immune-related gene expression in cancer cells such as TRAIL, TLRs, and FAS, regulates the STING pathway, influences the programmed death-ligand 1 (PD-L1) and supports cytotoxic T cell-induced tumor cell death resulting in modulation of anti-tumor immunity [35–37]. On the other hand, p53 can be transcriptionally activated by IFN after viral infections downstream of IFN signaling [30]. Therefore, we investigated the p53 transcriptional network in the presence and absence of IFN and identified its downregulation independent of IFN. To substantiate this finding, we infected p53-wildtype (A549) and p53-null (H1299) cell lines and observed downregulation of innate immunity associated genes in both. However, the extent of IFN-inducible gene repression was less pronounced in these tumor cell lines than in primary cells. Of note, primary fibroblasts (such as HFFs) and epithelial cancer cells (in our case, A549 and H1299 cells) differ in tissue origin and innate immune sensing. Because pattern recognition receptors vary by cell type and subcellular location, adenovirus entry routes can trigger distinct immune responses [38]. Thus, HAdV-C5 likely engages fibroblasts and epithelial cells differently, requiring distinct immune evasion strategies.

Virus progeny production was significantly reduced in the absence of E1B-55K, particularly in A549 cells, but not to the degree previously reported in HFFs, where yields of the ΔE1B-55K mutant were reduced at least 500-fold. Other studies have shown that IFN treatment of A549 cells has little impact on HAdV-C5 virus yield or genome accumulation [39,40], which is more consistent with our observations. Similarly, activated type I and III interferon responses negatively affected HAdV species mainly in primary epithelial cells, but not in A549 cells [41]. These results suggest that the full scope of E1B-55K-mediated transcriptional regulation of innate immunity may not be fully captured in tumor cell lines, despite robust IFN responses and induction of numerous ISGs (S3 and S4 Figs).

Transcriptional interference with immune response pathways and the functional outcome on genome replication and virus progeny was more pronounced in A549 upon IFN treatment. Therefore, we assumed that the intrusion of E1B-55K into these pathways is potentially facilitated via its interaction and inhibition of p53. This is especially important as increasing evidence shows that p53 also has a vital role in innate immunity during viral infections [22,37,42]. p53 has been implicated in antiviral responses, partly through regulation of IRF9 and interactions with other interferon regulatory factors such as IRF7 [30,43,44]. Despite shRNA-mediated knockdown of p53 in A549 cells, the levels of selected IFN-inducible proteins remained largely unchanged in time course experiments with wildtype and ΔE1B-55K mutant viruses. This may be due to compensatory mechanisms through p53-independent pathways. In general, accumulation

of p53 in the absence of E1B-55K was accompanied by comparable ISG expression. Based on our previous findings, this is unlikely due to additional viral proteins suppressing the response, as cells expressing only E1A and E1B also showed elevated p53 levels without induction of IFN signaling [19]. This suggests that p53 accumulation during adenoviral infection does not necessarily activate ISG expression or IFN responses, in contrast to other viruses. Whether E1B-55K represses ISG transcription directly or interferes indirectly – potentially via its SUMO E3 ligase activity [45,46] – remains to be clarified. Remarkably, E1B-55K expression during HAdV-C5 infection is not limited to repressing host transcription networks. We found that, in p53-positive A549 cells, upregulated genes were significantly enriched for pathways like oxidative phosphorylation, the mitochondrial electron transport chain, and cytoplasmic ribosomal proteins. Modulation of the cellular mitochondria-driven redox status presents an important strategy employed by many viruses to subvert host cellular pathways for their own benefit [47]. In p53-negative H1299 cells, E1B-55K-dependent upregulated genes were enriched for pathways involved in the host translation machinery, consistent with the role of E1B-55K in the accumulation and transport of late viral mRNAs as well as late viral protein synthesis [48–50]. To link the p53-positive and p53-negative contexts, it should be noted that HAdV-C5 replicates more efficiently in A549 cells (see virus yield and DNA replication, Figs 3C and S6), which may in turn require greater cellular resources. These findings suggest that E1B-55K may also influence cellular metabolism, potentially enhancing energy production and protein synthesis to support viral replication.

HAdVs, like polyoma- and papillomaviruses, are small DNA tumor viruses that disrupt host protein networks through viral hub proteins. We provide new data confirming that E1B-55K, similar to E1A, functions as such a hub – a role potentially conserved among (adeno-) viral oncogenes [5,6,51,52]. This study provides promising insights into the modulation of E1B-55K of host transcription factors during productive infection. E1B-55K promotes virus replication through inhibition of p53-mediated transcription and interference with type I IFN-mediated signaling, and others. Given known differences between adenovirus species, future comparative studies will be essential to fully define the broader role of E1B-55K in modulating antiviral host responses.

## Supporting information

**S1 Fig. Cell viability analysis.** (A) A549 and (B) H1299 cells were treated with universal IFN-α or PBS supplemented with 0.1% BSA (carrier) at indicated concentrations for 16-18h. After pre-treatment cells were infected with MOI 30 of wild-type (WT) or ΔE1B-55K virus. At 24 hpi, cell viability was measured using the CellTiter-Glo Assay (Promega) on the Spark multimode microplate reader (Tecan). Data points represent the mean of three technical replicates. Error bars indicate SD. The red dotted line marks 500 U/ml, which was used in the other assays. The horizontal dotted line marks the 75% cell viability threshold.
(TIF)

**S2 Fig. Complete pathway analysis from infected H1299 cells.** Wildtype and ΔE1B-55K infection at 24 hpi are compared in this analysis. Pathway analysis of up- and downregulated genes using GSEA, showing the normalized enrichment score of pathways with a FDR < 0.1 (light blue/red) and FDR < 0.05 (dark blue/red). Red bars indicate pathways that are enriched in upregulated genes, while blue bars indicate pathways that are enriched in downregulated genes. Databases utilized here were "Reactome", "KEGG", "WikiPathways" and "GO: Biological Process".
(TIF)

**S3 Fig. Robust activation of IFN-associated pathways following IFN treatment in A549 cells.** (A) Pathway analysis of up- and downregulated genes in mock-infected A549 cells comparing treatment with IFN-α or BSA using GSEA, showing the normalized enrichment score of pathways with a FDR < 0.1 (light blue/red) and FDR < 0.05 (dark blue/red). Pathways associated with the immune response are marked in pink. (B) Volcano plot of the comparison from (A). Genes with a padj. < 0.1 and $\log_2$ fold change < -1 were colored in blue, while genes with a padj. < 0.1 and $\log_2$ fold change > 1

were colored in red. Genes that belong to the "Interferon signaling" pathway from the Reactome database were additionally highlighted as dark red circles.
(TIF)

**S4 Fig. Robust activation of IFN-associated pathways following IFN treatment in H1299 cells.** (A) Pathway analysis of up- and downregulated genes in mock-infected H1299 cells comparing treatment with IFN-α or BSA using GSEA, showing the normalized enrichment score of pathways with a FDR<0.1 (light blue/red) and FDR<0.05 (dark blue/red). Pathways associated with the immune response are marked in pink. (B) Volcano plot of the comparison from (A). Genes with a padj.<0.1 and $\log_2$ fold change<-1 were colored in blue, while genes with a padj.<0.1 and $\log_2$ fold change>1 were colored in red. Genes that belong to the "Interferon signaling" pathway from the Reactome database were additionally highlighted as dark red circles.
(TIF)

**S5 Fig. Protein steady-state levels in sequenced biological replicates.** Western blot from sequenced replicates #2 and #3 of, displaying selected viral- and IFN-induced proteins as well as p53 target genes, as presented in Fig 3B. A549 (A) or H1299 (B) cells were treated with 500 U/ml IFN-α 16–18 h prior to infection with either wildtype or ΔE1B-55K virus. At 24 hpi, cells were harvested and RIPA extracts were prepared. For clarity, proteins associated with the p53 pathway are marked in green, while proteins associated with the immune response are marked in pink. The lower five proteins were visualized via the ChemoStar Plus, the upper ones were detected on medical X-ray films. Antibodies used for immunodetection can be found in S2 Table.
(TIF)

**S6 Fig. DNA replication assays.** (A) A549 and (B) H1299 cells were treated with universal IFN-α (500 U/ml) or PBS supplemented with 0.1% BSA (carrier) for 16–18 h. After pretreatment cells were infected with MOI 30 of wildtype (WT) or ΔE1B-55K virus and harvested at 2, 8 and 24 hpi. Relative amount of viral DNA was determined by qPCR using a L4-100K-specific primer pair (S1 Table). Ct values were normalized to GAPDH. DNA concentrations are plotted relative to 2 hpi. Bar graphs represent the mean of three biological replicates, error bars indicate SD. Black and purple bars represent wildtype and ΔE1B-55K virus, respectively. Statistical significance was determined using a two-tailed t-test. Significance levels are indicated as follows: ns>0.05.
(TIF)

**S7 Fig. Complete pathway analysis from IFN-treated infected A549 cells.** Wildtype and ΔE1B-55K infection at 24 hpi are compared in this analysis. Pathway analysis of up- and downregulated genes using GSEA, showing the normalized enrichment score of pathways with a FDR<0.1 (light blue/red) and FDR<0.05 (dark blue/red). The database utilized here is "WikiPathways".
(TIF)

**S8 Fig. Complete pathway analysis from IFN-treated infected H1299 cells.** Wildtype and ΔE1B-55K infection at 24 hpi are compared in this analysis. Pathway analysis of up- and downregulated genes using GSEA, showing the normalized enrichment score of pathways with a FDR<0.1 (light blue/red) and FDR<0.05 (dark blue/red). The database utilized here is "WikiPathways".
(TIF)

**S9 Fig. RT-qPCR validation of ISGs (pink) and p53 target gene (green) expression in A549 cells identified via RNA-seq.** The GAPDH gene is an internal control used for normalization. Boxplots show means of dCt values (with min to max values) of the three independent biological replicates used in NGS. Light grey, black, and purple box plots represent mock, wildtype virus, and ΔE1B-55K virus conditions, respectively. Primers used in RT-qPCR can be found in

S1 Table. Statistical significance was determined using a two-tailed t test. Significance levels are indicated as follows: * *P*-value < 0.05, ** *P*-value < 0.01, *** *P*-value < 0.001 and ns > 0.05.
(TIF)

**S10 Fig.  RT-qPCR validation of ISGs (pink) and p53 target gene (green) expression in H1299 cells identified via RNA-seq.** The GAPDH gene is an internal control used for normalization. Boxplots show means of dCt values (with min to max values) of the three independent biological replicates used in NGS. Light grey, black, and purple box plots represent mock, wildtype virus, and ΔE1B-55K virus conditions, respectively. Primers used in RT-qPCR can be found in S1 Table. Statistical significance was determined using a two-tailed t test. Significance levels are indicated as follows: * *P*-value < 0.05, ** *P*-value < 0.01, *** *P*-value < 0.001 and ns > 0.05.
(TIF)

**S11 Fig.  Effect of p53 knockdown on IFN response in A549 cells.** (A) IFN-α induction and time course scheme. (B) Western blot of IFN-treated (upper) or BSA-treated (lower) samples illustrating several adenovirus early and late proteins, as well as specific host cell p53- (green) and IFN-induced (pink) targets. Shown here is one replicate. The second replicate can be found in S12 Fig. Antibodies used for immunodetection can be found in S2 Table.
(TIF)

**S12 Fig.  Replicate analysis of IFN response following p53 knockdown in A549 cells.** Western blot of IFN-treated (upper) or BSA-treated (lower) samples illustrating several adenovirus early and late proteins, as well as specific host cell p53- (green) and IFN-induced (pink) targets. Shown here is one replicate. The first replicate can be found in S11 Fig. Antibodies used for immunodetection can be found in S2 Table.
(TIF)

**S13 Fig.  Impact of IFN treatment on adenovirus infection in HFF cells.** Western blot of three biological replicates, displaying selected viral-, p53-target (green) and IFN-induced (pink) proteins. HFF cells were treated with 500 U/ml IFN-α 16–18 h prior to infection with either wildtype or ΔE1B-55K virus. Cells were infected with MOI 100 and harvested at 48 hpi. Viral protein steady states are significantly reduced in the presence of IFN-α. Antibodies used for immunodetection can be found in S2 Table.
(TIF)

**S14 Fig.  RT-qPCR analysis of ISGs (pink) and p53 target gene (green) expression in HFF cells.** The GAPDH gene is an internal control used for normalization. Boxplots show means of dCt values (with min to max values) of the three independent biological replicates shown S13 Fig. Light grey, black, and purple box plots represent mock, wildtype virus, and ΔE1B-55K virus conditions, respectively. Primers used in RT-qPCR can be found in S1 Table. Statistical significance was determined using a two-tailed t-test. Significance levels are indicated as follows: * *P*-value < 0.05, and ns > 0.05.
(TIF)

**S1 Table.  Oligonucleotides.**
(DOCX)

**S2 Table.  Primary antibodies.**
(DOCX)

**S1 File.  Source data and statistical analysis.**
(XLSB)

## Acknowledgments

We thank all members of the Department of Viral Transformation and the LIV NGS technology platform for their support.

## Author contributions

**Conceptualization:** Laura Seddar, Konstantin von Stromberg, Luca D. Bertzbach, Thomas Dobner, Wing Hang Ip.

**Formal analysis:** Laura Seddar, Konstantin von Stromberg, Luca D. Bertzbach, Wing Hang Ip.

**Investigation:** Laura Seddar, Konstantin von Stromberg, Wing Hang Ip.

**Project administration:** Laura Seddar.

**Supervision:** Adam Grundhoff, Thomas Dobner, Wing Hang Ip.

**Visualization:** Laura Seddar, Konstantin von Stromberg.

**Writing – original draft:** Laura Seddar, Konstantin von Stromberg, Luca D. Bertzbach.

**Writing – review & editing:** Laura Seddar, Konstantin von Stromberg, Luca D. Bertzbach, Adam Grundhoff, Wing Hang Ip.

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
