## [Decision Letter · Decision Letter 0]

9 Aug 2025

PPATHOGENS-D-25-01571

Adenovirus E1B-55K regulates p53-dependent and -independent gene expression during infection

PLOS Pathogens

Dear Dr. Ip,

Thank you for submitting your manuscript to PLOS Pathogens. After careful consideration, we feel that it has merit but does not fully meet PLOS Pathogens's publication criteria as it currently stands. Therefore, we invite you to submit a revised version of the manuscript that addresses the points raised during the review process.

Please submit your revised manuscript within 30 days Oct 08 2025 11:59PM. If you will need more time than this to complete your revisions, please reply to this message or contact the journal office at plospathogens@plos.org. Please include the following items when submitting your revised manuscript:

We look forward to receiving your revised manuscript.

Kind regards,

Corey Smith

Academic Editor

PLOS Pathogens

Donna Neumann

Section Editor

PLOS Pathogens

Sumita Bhaduri-McIntosh

Editor-in-Chief

PLOS Pathogens

orcid.org/0000-0003-2946-9497

Michael Malim

Editor-in-Chief

PLOS Pathogens

orcid.org/0000-0002-7699-2064

**Additional Editor Comments:**

Please response appropriately to all of the comments provided by the reviewers, particularly taking note of the major issue raised by reviewer 1 and the corrections suggested by reviewers 1 and 3.

**Journal Requirements:**

2) Your paper currently exceeds the word limit of 1,800 words for this article type. Please either revise your submission to be within the word limit or change the article type to Research Article.

4) Please amend your detailed Financial Disclosure statement. This is published with the article. It must therefore be completed in full sentences and contain the exact wording you wish to be published.

**Reviewers' Comments:**

Reviewer's Responses to Questions

**Part I - Summary**

Reviewer #1: This short report examines the function of the human adenovirus type 5 E1B-55K protein in regulating host gene expression during the infection of two cell lines (A549 and H1299) using a mutant virus lacking E1B-55K and its parental strain. Transcriptomic and protein-based analyses confirm that E1B-55K suppresses p53-dependent transcription, a well-studied property which has primarily been investigated outside the context of adenovirus infection. Furthermore, E1B-55K was found to modulate interferon (IFN)-stimulated gene expression in A549 cells, but not in H1299 cells.

This study will be of great interest to the field. The results are clearly presented, and the conclusions are generally justified. However, the following issues need to be addressed.

Reviewer #2: The manuscript by Seddar et al studies the Adenovirus E1B-55K protein. This protein is best known as an oncogene that targets the p53 tumor suppressor. In addition, 55K also forms a complex with a second viral protein (E4orf6) and targets key cellular substrates for degradation including p53 and the Mre11 complex. In the current study, the authors show a previously unknow role for 55K in the regulation of interferon (IFN) responses.

The authors use transcriptomics approaches to study the differences in gene expression between wild-type Ad5 and mutant virus lacking 55K during infection of A549 and H1299 cells. As expected, the major differences are in the p53 pathway. What was less expected was that changes in IFN dependent gene expression were also observed. To follow up these observations, the authors go on to show that the E1B-55K null mutant is more sensitive to IFN pre-treatment. Since the p53 pathway is known to have cross talk with the IFN pathway, the conducted similar experiments in A549 cells in which p53 is knocked down. Under these conditions, IFN dependent gene expression is still suppressed suggesting that the mechanism is independent of p53.

Overall, the manuscript is an excellent contribution to our understanding of Ad E1B-55K and will be of general interest to DNA tumor virologists. The big remaining question is the mechanism by which 55K mediates its effects on IFN gene expression. Most likely, there is an unknow target of the orf6/55K complex that mediates these effects?

Reviewer #3: Seddar et al. have performed a comprehensive RNA-seq based study of the role of E1B-55K during adenovirus infection, focusing on it’s role in controlling p53 signaling and in response to IFN treatment. The study is interesting and informative as previous papers tend to focus on E1B-55K’s role in transformation. The manuscript, however, suffer for some lack of details in many aspects and because of that can be more difficult to read and understand.

For example, the authors have generated RNA-seq data on many different sets of conditions, including mock, WT and 55K deleted infections; H1299 and A549 cell lines; control and IFN treated cells as well as some combinations. It is not always clear which comparison is shown, particularly in volcano plots and to which condition increases or decreases in reads refer to. In other cases, no description is provided in the text on how the experiments were conducted. Part of the issue may be due to space limitation for a short report. It might be worth considering a normal format to fully address the all the implication of the work

**Part II – Major Issues: Key Experiments Required for Acceptance**

Reviewer #1: 1. The authors must validate any key differential changes identified in their RNA-seq analyses using RT-qPCR or a similar RNA-based method. Alternatively, they must justify why they believe such validation is unnecessary.

Reviewer #2: No major issues.

Reviewer #3: no major issues

**Part III – Minor Issues: Editorial and Data Presentation Modifications**

Reviewer #1: 2. The authors should explain why they chose the A549 and H1299 cell lines for this study. Cancer cell lines often exhibit altered IFN and p53 responses. Apart from the fact that the latter are p53-negative, what is known about the IFN and p53 responses in A549 and H1299 cells? The data should be discussed in this context.

3. An imbalance exists between the data presented in the main manuscript (3 figures) and that included in the supplement (2 tables and 11 figures, as well as additional methods). It is unclear why the authors decided to present their work as a short report rather than a full-length one, which would have allowed more figures to be included in the main manuscript.

4. Figure 1A: The text should mention that reduced levels of Mre11, p53 and CDKN1A were not observed in the E1B-55K mutant virus.

5. Figure 1D: To say that 'these genes remained transcriptionally active in both mock-infected and ΔE1B-55K-infected cells' is an overstatement, since many p53-responsive genes are downregulated in cells infected with either the wild-type or mutant viruses, albeit to different extents.

6. The regulation of pathways other than those dependent on p53 and IFN by E1B-55K should be discussed, including its potential effects on metabolism, as illustrated in Figure 1E.

7. The conclusions that 'infection with adenovirus also affected the TEAD transcriptional network' (line 143) and 'our analysis revealed consistent downregulation of the Hippo signaling pathway' (line 144) appear to apply to infected H1299 cells, but not to A549 cells. This should be clearly stated and discussed in the text. Additionally, the text should specify which TEAD protein (TEAD4) was investigated for physical interaction with E1B-55K in Figure S2.

8. The authors should acknowledge that the IFN-related differences they observe between wild-type and mutant viruses, as well as between A549 and H1299 cells (Figures 2 and 3), tend to be small. It is debatable whether some of these effects are genuine or merely spurious (e.g. differences in viral DNA replication between A549 and H1299 cells in Figure 2C).

9. Although the manuscript is written well overall, it should be carefully checked for small spelling errors and inconsistencies. These include, but are not limited to, 'by targeting *a* host tumour suppressor' (Abstract) and 'displaying *raw* Z-scores' (Figure 1D legend).

10. The text shows signs of potential AI (LLM) use. If the authors have used AI in the writing process, they should acknowledge this.

Reviewer #2: The manuscript is very well written and referenced. The data is nicely presented with all relevant statistics provided. The only suggestion I can make for improvement is that the molecular weight markers in supplemental figures 8 and 9 are missing. The authors should include these as they have with the other immunoblots in the manuscript.

Reviewer #3: Detailed issues

- In the results section, the description of Figure 1 starts very abruptly without details of the experiment. The first sentence starts with: “Western blot analysis confirmed productive infection …”. Also the data in fig. 1A does not actually confirm a productive infection as it is only showing expression of 55K, some substrates of the orf6/55K ligase complex, and down stream targets of p53. The wording should be modified to be more precise.

- Fig. 1E are red (increased) lines indicate increased in mutant infected cells relative to the wt infected cells? This should be clearly indicated.

- Line #143-147. No details of what the experiment was or cell line. The text suggests WT infection compared to no infection, however, the Figure legend for S2 seems to indicate WT infection compared to mutant virus infection. Should be clarified.

- Fig. S3 “Pathway analysis of up- and downregulated genes” Need to precise what is being compared here.

- Line #172-173 The wording in incorrect. “FN-α exacerbated the replication defect 173 of the ΔE1B-55K mutant, particularly in H1299 cells (Figs. 2C and 3A).” In 2C it only shows that there is a replication defect in A549 cells while panel 3A does not show any replication data. Need to be changed.

- Which data set is shown in Fig 3B? it is not clear.

More minor issues and typos

- Fig. 1C The text says that p53 target genes are marked in green. At leas on my screen all dots look black

- Fig. 1D The grey, black and purple boxes shown on the top of the panel probably indicate the same as in panel B, but that should be noted in the figure legend.

- Supplementary Figure 1 is not mentioned in the text. It should be or otherwise removed.

- Line #143 “infection with adenovirus also affected the TEAD transcriptional network” SF2 should be indicated here

- Fig S2 legend title mentions in absence of p53, but the cell line used, H1299 should be mentioned at the beginning, not just for panel E. The letters for panel C, D and E should be in bold like A and B.

- Fig S2C refers to Fig. S2. That should be Fig. S3

- Figure legend for Fig. S2E. “expression of transfected E1B-55K and exogenous TEAD4” What does exogenous means here? It appears to also be transfected.

- Fig. 2D should specify in the legend what the white and pink banding means. It’s on the figure, but not obvious and hard to see.

- Line #160. Should be Fig 2B and not 3B

- Line #166 the 24 post infection should be 24h

- SF7 legend. Only need to describe the 1 star as there is only that condition in the figure

- Line #178 “ISG expression was consistently lower in infected cells than in the uninfected” That is not the case for all.

- Figure legend for panel 3D need to indicate which cell line is on the left and right.

PLOS authors have the option to publish the peer review history of their article (what does this mean? ). If published, this will include your full peer review and any attached files.

**Do you want your identity to be public for this peer review?** For information about this choice, including consent withdrawal, please see our Privacy Policy .

Reviewer #1: No

Reviewer #2: No

Reviewer #3: No

**Figure resubmission:**
---

## [Decision Letter · Decision Letter 1]

13 Oct 2025

Dear Dr Ip,

We are pleased to inform you that your manuscript 'Adenovirus E1B-55K regulates p53-dependent and -independent gene expression during infection' has been provisionally accepted for publication in PLOS Pathogens.

Best regards,

Corey Smith

Academic Editor

PLOS Pathogens

Donna Neumann

Section Editor

PLOS Pathogens

Sumita Bhaduri-McIntosh

Editor-in-Chief

PLOS Pathogens

orcid.org/0000-0003-2946-9497

Michael Malim

Editor-in-Chief

PLOS Pathogens

orcid.org/0000-0002-7699-2064

Reviewer Comments (if any, and for reference):

Reviewer's Responses to Questions

**Part I - Summary**

Reviewer #1: (No Response)

Reviewer #2: The authors have diligently responded to the reviewer comments. I have no further critiques.

Reviewer #3: The authors have addressed my previous comments in a satisfactory manner. I now believe that the manuscript is suitable for publication

**Part II – Major Issues: Key Experiments Required for Acceptance**

Reviewer #1: (No Response)

Reviewer #2: none

Reviewer #3: none

**Part III – Minor Issues: Editorial and Data Presentation Modifications**

Reviewer #1: (No Response)

Reviewer #2: none

Reviewer #3: none

PLOS authors have the option to publish the peer review history of their article (what does this mean? ). If published, this will include your full peer review and any attached files.

**Do you want your identity to be public for this peer review?** For information about this choice, including consent withdrawal, please see our Privacy Policy .

Reviewer #1: No

Reviewer #2: No

Reviewer #3: No

---

## [Editor Report · Acceptance letter]

Dear Mrs. Ip,

We are delighted to inform you that your manuscript, "Adenovirus E1B-55K regulates p53-dependent and -independent gene expression during infection," has been formally accepted for publication in PLOS Pathogens.

Best regards,

Sumita Bhaduri-McIntosh

Editor-in-Chief

PLOS Pathogens

orcid.org/0000-0003-2946-9497

Michael Malim

Editor-in-Chief

PLOS Pathogens

orcid.org/0000-0002-7699-2064